# DDL: Dynamic Discard Deep Learning for Rice Yield Prediction on Mixed-Accuracy Datasets

## Abstract

Effectively fusing scarce high-accuracy data with massive but noisy low-accuracy data is a common challenge faced by machine learning across various fields, including agriculture, medicine, and remote sensing. Existing methods, which either directly concatenate datasets while ignoring accuracy differences or employ static weighting for training, struggle to achieve optimal performance. To address this, we introduce a deep learning framework incorporating a dynamic discard mechanism (DDL) that manages mixed-accuracy data through the selective, dynamic removal of low-accuracy instances characterized by high Mean Absolute Error (MAE) and the application of an adaptive weighting scheme. Our study validated this approach using rice cultivation data from China's four major rice-growing regions: South, Central, North, and Northeast China. Using site characteristics and nitrogen application rates as feature variables and rice yield as the target variable, we designated the high-accuracy dataset as the test set. Compared to machine learning models that process only single-accuracy datasets and other models designed for mixed-accuracy data, our DDL framework demonstrated a performance improvement of over 10% in metrics such as RMSE, MAE, and MAPE, achieving significantly higher prediction accuracy. A crop yield prediction model capable of handling multiple datasets simultaneously holds significant practical value for policymakers and other stakeholders. The dynamic discard mechanism and adaptive weighting algorithm employed by DDL also have considerable reference value for applications in other domains.

## 1 Introduction

Accurate estimation of crop yields is crucial for ensuring global food security and maintaining a stable market economy (Basso & Liu, 2019; Lecerf et al., 2019). This is fundamental for developing reasonable agricultural policies and effective food management (Zambrano et al., 2018), and also promotes sustainable agricultural development. In genuine agricultural contexts, crop productivity is synergistically governed by an intricate confluence of diverse parameters, notably edaphic and climatic factors, as well as nutrient application rates(Lai et al., 2024; Shuai & Basso, 2022). The intricate nature of these environmental variables makes it challenging to acquire a sufficient amount of high-accuracy data, which constrains the improvement of model prediction accuracy and generalization ability. In this context, building a prediction model that can effectively integrate datasets of mixed accuracy has become the core approach to enhancing estimation accuracy and generalization capability.

Agricultural data obtained directly from field observations and experiments are highly accurate and reliable but are limited in sample size due to high costs and labor-intensive processes. While existing research has used various machine learning (ML) models and deep neural networks (DNNs) to estimate crop yields and improve prediction accuracy (Akkem et al., 2023; Han et al., 2025). While most of these studies rely on data augmentation to process existing datasets for model training. This paucity of adequate sample diversity invariably compromises model generalizability, thereby failing to ameliorate the intrinsic data scarcity bottleneck inherent in agricultural domains.

In contrast, data generated through existing agricultural process models offer a massive sample size but are of lower accuracy. The simultaneous existence of high- and low-accuracy data presents both a significant challenge and a potential solution to agricultural data scarcity. However, the effective

fusion of scarce high-accuracy data with vast but noisy low-accuracy data remains an unresolved problem. Most existing studies either directly concatenate datasets, thereby overlooking disparities in data accuracy, and attempt to mitigate the noise introduced by lower-accuracy data through preprocessing methods (Zhang et al., 2022a). However, these approaches struggle to suppress the negative impact on model training while simultaneously preserving the intrinsic information content of the data. Alternatively, some studies employ static weighting strategies to enhance the fusion effect of mixed-accuracy data (Gao & Xie, 2025), yet these methods inherently lack adaptability to the dynamic changes occurring throughout the model training process.

To address these issues, this study proposes and validates a deep learning framework based on a dynamic discard mechanism (DDL), which can train datasets of different accuracies simultaneously. By leveraging a dynamic discard algorithm and an adaptive weighting mechanism, the framework enhances the model's prediction accuracy and generalization ability, effectively solving the mixed-accuracy fusion problem.

The remainder of this paper is organized as follows: Section 2 reviews related work in this field; Section 3 describes the data acquisition strategies and sources; Section 4 presents the DDL architectural design and various mechanisms; Section 5 shows the model results; and Section 6 discusses the research findings and outlines future research directions.

## 2 RELATED WORK

Recent research has increasingly focused on strategies to address the challenges of integrating heterogeneous or uncertain datasets into predictive modeling.

**CNN–GAN–based methods** Methods combining Convolutional Neural Networks (CNNs) and Generative Adversarial Networks (GANs) have been widely used to improve data quality and augment training samples. By learning high-level feature representations and generating realistic synthetic samples, these approaches effectively mitigate the issue of insufficient training data and enhance model generalization. However, they typically assume a homogeneous data accuracy, often performing poorly when there are systematic differences in the reliability of training samples(Zhang et al., 2022b).

**U-Net with ConvLSTM architectures** To capture spatiotemporal dependencies in agricultural and environmental applications, researchers have proposed hybrid models that fuse the U-Net architecture with Convolutional Long Short-Term Memory (ConvLSTM) modules. These models effectively integrate sequential information with spatial context, leading to significant improvements in tasks like crop monitoring. Nevertheless, their fusion process relies on a static architecture, lacking a dynamic mechanism to adapt to variations in input data quality(Kamangir et al., 2025).

**Remote sensing data assimilation with SCE-UA** Another research path involves data assimilation methods, such as utilizing optimization algorithms like the Shuffled Complex Evolution Algorithm (SCE-UA) to combine remote sensing observations with process-driven models. These approaches explicitly merge observations and simulated values, and can significantly enhance predictive accuracy, particularly when observations are sparse or noisy. However, they typically operate within a deterministic optimization framework, lacking a mechanism to adaptively discard or re-weight low-accuracy samples during training(Li et al., 2024).

**Dynamic Reweighting and Data Selection** While our work shares the goal of improving training dynamics with methods like Population Based Augmentation (PBA) (Ho et al., 2019) and Sample Reweighting (Ren et al., 2018), there is a fundamental distinction necessitated by the nature of mixed-accuracy scientific data. Traditional reweighting methods assign soft weights to high-loss samples, effectively down-weighting outliers but retaining them in the optimization process. In the context of simulations, high-error samples often represent systematic failures rather than aleatoric noise. Retaining these samples, even with low weights, risks corrupting the feature manifold.

In contrast, DDL employs a hard Dynamic Discard mechanism. By completely removing samples that persistently diverge from the high-accuracy distribution, DDL prevents the model from fitting to systematic biases, offering a more robust solution for integrating heterogeneous scientific datasets than augmentation or static reweighting strategies.

## 3 DATA STRATEGY AND SOURCES

The primary objective of this study is to enhance the predictive accuracy and generalization capability of the model to the greatest extent possible. However, during data collection, we identified a pervasive challenge: the quality and quantity of the required data exhibit an inverse relationship, resulting in two distinct types of data sources. To address this challenge, we propose an integrative strategy that combines the use of both high-accuracy and low-accuracy data.

The central premise of our data strategy is to combine the complementary strengths of these two sources—namely, the precision of high-accuracy data and the breadth of low-accuracy data—to construct a more robust and high-performing predictive model.

### 3.1 HIGH-ACCURACY DATA: COLLECTED EMPIRICAL OBSERVATIONS

We established a high-accuracy, field-scale rice yield dataset through systematic field observations conducted between 2021 and 2024 across four major rice-growing regions. All data were collected under standardized experimental protocols with rigorous quality control procedures, ensuring consistency and reliability. In total, 704 observational records were obtained, providing a comprehensive empirical basis for subsequent modeling and analysis.

### 3.2 LOW-ACCURACY DATA: DNDC-BASED SIMULATIONS

The DNDC (DeNitrification-DeComposition) model (Li et al., 1992) is a process-based biogeochemical model of carbon and nitrogen dynamics in agroecosystems. By coupling microbial metabolic processes with the soil's physical environment, DNDC enables refined simulations of C–N cycles in complex agricultural systems. In this study, we employed version 9.5 of the DNDC model to simulate crop planting from paddy fields.

Soil property data and climate data obtained from the National Meteorological Science Data Center (2024) were aggregated at a 0.5° resolution into a format compatible with DNDC input requirements. We then ran the DNDC model in Region Mode, the simulation outputs include process-level crop growth data. From these results, we selected key variables including latitude/longitude, SOC, clay content, pH, bulk density (BD), average temperature, precipitation, irrigation, nitrogen application, crop type, and yield. In total, we obtained 43447 simulated records.

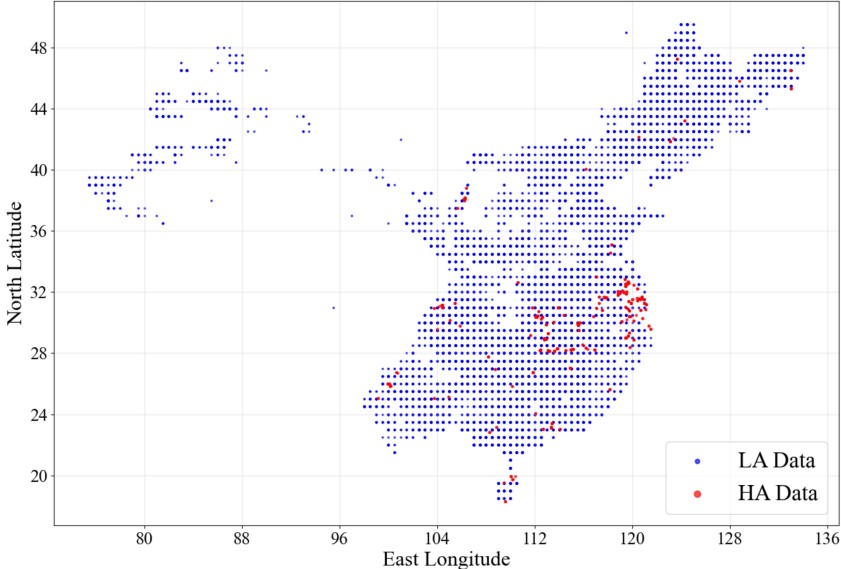

Figure 1: Mixed-accuracy datasets distribution

# 4 METHODS

## 4.1 NOTATION AND PROBLEM SETTING

This study addresses the problem of joint prediction by leveraging a small-sample, high-accuracy dataset and a large-sample, low-accuracy dataset. Let $D^H = \{(x_i^H, y_i^H)\}_{i=1}^{n_H}$ and $D^L = \{(x_i^L, y_i^L)\}_{i=1}^{n_L}$ denote the high-accuracy (collected) data and the low-accuracy (simulated) data respectively. Here, $x_i$ represents an 11-dimensional feature vector: Longitude, Latitude, Tem (temperature), Pre (precipitation), rate (fertilizer application N rate), SOC (soil organic carbon), Clay (clay content), pH (soil pH), BD (soil bulk density), irrigation, and CropType and $y_i$ is the target variable, **Yield** (rice yield per mu). The dataset sizes are $n_H = 704$ and $n_L = 43,447$. Our primary objective is to achieve superior predictive accuracy on a test set partitioned from the high-accuracy dataset, while also maximizing the model's generalization capability. In our pipeline the high-accuracy dataset was split prior to model training: 20% of $D^H$ were reserved as a final, held-out test set, 20% of the remaining high-accuracy samples were used as a validation set, and the remainder were used for training (all random splits used seed 42). The two datasets therefore share the same feature / label schema but differ in measurement accuracy and noise characteristics. The held-out $D^H$ test subset was not used in any way during training, dynamic discard procedures, or adaptive weight tuning.

## 4.2 ARCHITECTURE AND TECHNICAL DESIGN OF THE DDL FRAMEWORK

As illustrated in Figure 1, we propose the DDL architecture, which consists of an input layer, a **Feature Attention Gating Module**, a **Modified Residual Block**, a fully connected layer, and a multi-task output layer. The model incorporates a **dynamic dropout mechanism** and **adaptive dynamic weights**. The specific technical principles of these components are detailed below.

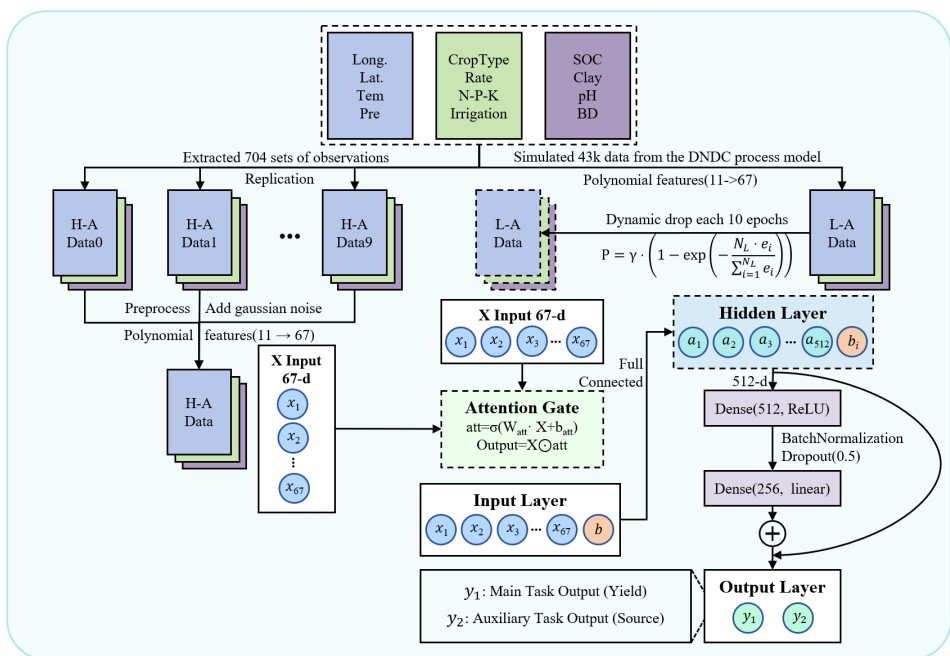

Figure 2: Schematic overview of the DDL framework. The architecture comprises a Feature Attention Gating Module for input reweighting, Modified Residual Blocks to mitigate gradient vanishing, and a multi-task output layer. The Dynamic Discard mechanism and Adaptive Dynamic Weights operate during training to filter noise and balance task losses.

### 4.2.1 DYNAMIC DISCARD STRATEGY

The core mechanism of our model is the **Dynamic Discard (DD) strategy**, which is not a static data cleaning process performed before training. Instead, it is a progressive filtering mechanism that evaluates and discards low-accuracy data samples during the training process based on the model's current predictive performance, thereby gradually filtering out noisy data.

Specifically, at epoch $t$, with current model parameters $\theta_t$, we first perform a forward pass on all low-accuracy data samples in $D^L = \{(x_i^L, y_i^L)\}_{i=1}^{N_L}$ to obtain their primary task (yield prediction) predictions, $\widehat{y_i^L} = f_{\theta_t}(x_i^L)$. We then calculate the absolute prediction error $e_i = |\hat{y}_i^L - y_i^L|$ for each low-accuracy sample.

To ensure the discard threshold adapts to the model's performance at different training stages, we compute the Mean Absolute Error (MAE) (Hodson, 2022) of all low-accuracy samples in the current batch as a dynamic baseline:

$$MAE_t = \frac{1}{n_L} \sum_{i=1}^{n_L} |\hat{y}_i^L - y_i^L|$$

Next, we normalize each sample's error to obtain a deviation metric relative to the model's current average performance: $\widetilde{e}_i = e_i / \text{MAE}_t$.

Based on this deviation, we calculate a dynamic discard probability $P_i$ for each sample, determined by an exponential decay function:

$$P_i = \gamma \cdot (1 - \exp(-\widetilde{e}_i))$$

where $\gamma$ is a base discard probability factor that controls the steepness of the probability curve, thereby regulating the penalty on high-error samples. Finally, a Bernoulli sampling process (Yu et al., 2022) determines whether the sample is kept: a random number $r$ is generated in the range $[0, 1]$. If $r < P_i$, the sample is discarded; otherwise, it is retained for the current and subsequent gradient updates.

This process can be conceptualized as a filtering operator $\mathcal{D}_t$ that acts on the low-accuracy dataset, outputting a filtered subset:

$$\mathcal{D}_t(\{(\mathbf{x}_i^L, y_i^L)\}) = \{(\mathbf{x}_j^L, y_j^L) \mid r_j \geq p_j\}$$

This subset is then combined with the high-accuracy data $D^H$ to form the training data for epoch $t$. Through this mechanism, the model can initially leverage the large volume of low-accuracy data to quickly learn general features. With advancing training and enhanced model performance, the DD mechanism adopts a more stringent approach, systematically filtering out low-quality samples that persistently yield high errors and are likely indicative of noise or substantial divergence from the true distribution. This allows the model to later focus on refining its predictive capabilities using higher-quality data, effectively preventing the negative influence of low-quality data and achieving a dynamic balance between data quality and quantity.

In summary, the dynamic discard probability formula of the proposed model is:

$$P_i = \gamma \cdot \left(1 - \exp\left(\frac{-N_L \cdot e_i}{\sum_{j=1}^{N_L} e_j}\right)\right)$$

### 4.2.2 FEATURE ATTENTION GATING MODULE AND REGULARIZATION

The model's input layer receives 11 types of features, including soil parameters, meteorological factors, and crop data. To enhance the importance of features highly correlated with the target variable, we designed a trainable **Feature Attention Gating Module** (Meng et al., 2022; Dhingra et al., 2016). This module uses fully connected layers (Basha et al., 2020) to perform dynamic feature weighting at the input layer, unlike the more computationally expensive self-attention mechanism, ensuring core features play a primary role in the final prediction.

The mathematical formulation is as follows:

$$\text{att} = \sigma(W_{\text{att}} \cdot x + b_{\text{att}})$$

$$\text{output} = x \odot \text{att}$$

where $\sigma$ is the attention gate vector, $W_{\text{att}}$ and $b_{\text{att}}$ are trainable parameters, and $\odot$ denotes element-wise multiplication.

To prevent model overfitting, we introduce L2 regularization (Van Laarhoven, 2017), which modifies the total objective function to:

$$\mathcal{L}_{reg} = \mathcal{L} + \lambda \, ||W_{\text{att}}||_F^2$$

where $\lambda = 10^{-4}$ is the regularization coefficient and $|| \cdot ||_F$ is the Frobenius norm of the weight matrices. This regularization term can be interpreted as a Gaussian prior on the weight parameters, ensuring that the attention weights do not become overly concentrated on a few features.

### 4.2.3 MODIFIED RESIDUAL BLOCK

To address the issue of **gradient vanishing** in deep networks (Tan & Lim, 2019), we designed a modified residual block structure (Zhang et al., 2017; Tang et al., 2024). Each block consists of two fully connected layers: the first employs a **ReLU** activation function (He et al., 2018)to introduce non-linearity, while the second uses a linear activation to maintain numerical stability. A skip connection then adds the block's input to its output.

The mathematical expression is as follows:

$$x^{(l+1)} = g\left(x^{(l)}\right) + \mathcal{F}\left(x^{(l)}\right)$$

where $g\left(x^{(l)}\right)$ is a dimension-adapting function that acts as an identity mapping when the input dimension $\dim\left(x^{(l)}\right)$ matches the target output dimension $d_{\text{out}}$ ; otherwise, it performs a dimensionality transformation via a projection matrix $W_s$ :

$$g\left(x^{(l)}\right) = \begin{cases} x^{(l)}, & \text{if } \dim\left(x^{(l)}\right) = d_{\text{out}} \\ W_s x^{(l)} + b_s, & \text{otherwise} \end{cases}$$

The residual function $\mathcal{F}\left(x^{(l)}\right)$ is implemented with two fully connected layers, and its computational flow is given by the following equation:

$$\mathcal{F}\left(x^{(l)}\right) = W_2^{(l)} \cdot z_3 + b_2^{(l)}$$

where $z_3$ is an intermediate variable computed as follows:

$$z_1 = \delta_{\text{ReLU}}\left(W_1^{(l)} x^{(l)} + b_1^{(l)}\right)$$

$$z_2 = \text{BN}\left(z_1\right)$$

$$z_3 = \text{Dropout}\left(z_2\right)$$

Here, $\delta_{\text{ReLU}}$ denotes the ReLU activation function. We constructed these identity mapping paths within the 512- and 256-dimensional hidden layers. This allows gradients to be passed directly to the shallower layers during backpropagation, effectively mitigating the vanishing gradient problem. Combined with **Dropout**, this structure also provides a strong regularization effect.

### 4.2.4 ADAPTIVE DYNAMIC WEIGHTS

To enable the model to learn high-accuracy predictions from the small dataset while simultaneously improving its generalization from the large dataset, we designed a framework with **adaptive dynamic weights** (Yang et al., 2022; Xiao & Zhang, 2021; Cao et al., 2023). This approach dynamically balances the loss weights between the main task (regression prediction) and an auxiliary task (data source classification).

The multi-task learning framework consists of a main task (predicting the target variable via regression) and an auxiliary task (predicting the data source as a binary classification task). The overall loss function is defined as:

$$\mathcal{L}_{total} = \alpha \cdot \mathcal{L}_{main} + (1 - \alpha) \cdot \mathcal{L}_{aux}$$

where $\alpha$ is a dynamic weighting coefficient, $\mathcal{L}_{\text{main}}$ is the mean squared error (MSE) loss, and $\mathcal{L}_{\text{aux}}$ is the classification cross-entropy loss:

$$L_{main} = \frac{1}{N} \sum_{i=1}^{N} (y_i - \hat{y}_i)^2$$

$$\mathcal{L}_{aux} = -\frac{1}{N} \sum_{i=1}^{N} [s_i \log(\hat{s_i}) + (1 - s_i) \log(1 - \hat{s_i})]$$

The initial value of $\alpha$ is set to 0.4 and is increased exponentially with each epoch, up to a specified limit. This mechanism ensures that the model initially leverages the classification task to enhance its generalization, then later focuses on optimizing predictive accuracy as the main task's influence increases. The dynamic weight update mechanism is as follows:

$$\alpha_{\text{epoch}+1} = \max(0.4, \min(0.8, \alpha_{\text{epoch}} \times 1.005))$$

## 5 RESULT

### 5.1 EXPERIMENTAL RESULTS AND DISCUSSION

The experimental results demonstrate that the proposed DDL framework significantly outperforms existing mainstream methods for the rice yield prediction task. As shown in Table 1, DDL exhibits superior performance across all four metrics: MAE, RMSE, R², and MAPE, surpassing traditional machine learning models (e.g., XGBoost, Random Forest), process-based models (e.g., DNDC), and domain-specific hybrid architectures (e.g., CNN+GAN, Remote Sensing Assimilation + SCE-UA). The framework achieves an approximate 10% improvement on these metrics, with the predictive R² value reaching 0.68. This strong agreement between predicted and observed yields is further visualized in the scatter plot of Figure 3, where most samples cluster closely around the fitted regression line, particularly in the mid-to-high yield range.

Table 1: Performance Comparison of Various Models on Regression Tasks

| Model | MAE $\downarrow$ | RMSE $\downarrow$ | $R^2 \uparrow$ | MAPE $\downarrow$ | Reference |
|---|---|---|---|---|---|
| **ML Model** | | | | | |
| Gradient Boosting R | 1001.96 | 1287.90 | 0.6348 | 0.1648 | (Friedman, 2001) |
| LightGBM | 1011.40 | 1316.29 | 0.6186 | 0.1678 | (Ke et al., 2017) |
| Random Forest | 1076.34 | 1436.98 | 0.5454 | 0.1754 | (Breiman, 2001) |
| XGBoost Regression | 1064.40 | 1388.34 | 0.5757 | 0.1750 | (Chen & Guestrin, 2016) |
| GPR | 1296.37 | 1808.19 | 0.3381 | 0.2098 | (Rasmussen & Williams, 2006) |
| MHA-MLP | 1115.07 | 1496.70 | 0.5465 | 0.1767 | - |
| **Related Work** | | | | | |
| CNN+GAN | 1115.56 | 1637.84 | 0.3587 | 0.2026 | (Zhang et al., 2022b) |
| UNet-ConvLSTM | 1075.91 | 1357.26 | 0.5497 | 0.1672 | (Kamangir et al., 2025) |
| PBA-ResNet | 1082.35 | 1462.31 | 0.4823 | 0.1612 | (He et al., 2016) |
| PBA-MLP | 1108.91 | 1642.98 | 0.4535 | 0.1635 | (Ho et al., 2019) |
| Broad Learning System | 1160.32 | 1559.85 | 0.5074 | 0.1815 | (Liu & Chen, 2018) |
| **Process Model** | | | | | |
| DNDC Model | 1439.35 | 1754.55 | 0.2910 | 8.7361 | (Li et al., 1992) |
| **DDL Model** | | | | | |
| Ours | **852.30** | **1212.14** | **0.6837** | **0.1424** | - |

### 5.2 ABLATION STUDY

An ablation study was conducted to validate the necessity and effectiveness of each component within the DDL framework. The results confirm that the absence of any single component leads to a decline in model performance. Specifically, a "single-accuracy" version of the model trained exclusively on the high-accuracy data demonstrated significantly limited performance. Similarly, models that did not employ the **dynamic discard mechanism** during mixed-precision training or those that used a static weighting strategy in place of the **adaptive dynamic weights** both performed

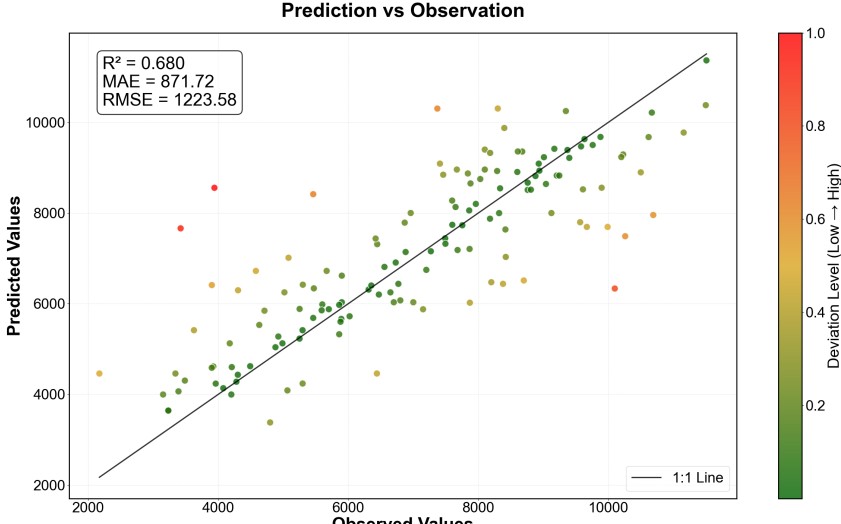

Figure 3: Scatter plot of predicted versus observed rice yield values. Each point corresponds to a sample, with its color indicating the absolute deviation from the fitted regression line: green denotes close agreement (small error), gradually transitioning to red for larger deviations (poor agreement)

worse than the complete framework. These findings in Table 2 suggest that static data fusion or the unfiltered use of low-accuracy data fails to effectively suppress noise interference. The synergistic design of the **dynamic discard** and **adaptive dynamic weights** is, therefore, crucial for DDL's ability to achieve high-precision predictions. The results of the ablation study fully validate the necessity and efficacy of the proposed mechanisms in fusing mixed-precision data.

Table 2: Ablation Study of the DDL Framework

| Model | MAE ↓ | RMSE ↓ | $R^2$ ↑ | MAPE ↓ |
|---|---|---|---|---|
| **ML Model** | | | | |
| Gradient Boosting R | 1001.96 | 1287.90 | 0.6348 | 0.1648 |
| **Related Work** | | | | |
| UNet-ConvLSTM | 1075.91 | 1357.26 | 0.5497 | 0.1672 |
| **Process Model** | | | | |
| DNDC Model | 1439.35 | 1754.55 | 0.2910 | 8.7361 |
| **DDL Model** | | | | |
| Ours(High-Accuracy Only) | 958.62 | 1335.92 | 0.6158 | 0.1602 |
| Ours(w/o Dynamic Discard) | 887.10 | 1273.89 | 0.6506 | 0.1477 |
| Ours(w/ Static Weighting) | 870.31 | 1249.90 | 0.6637 | 0.1439 |
| Ours(Full) | **852.30** | **1212.14** | **0.6837** | **0.1424** |

## 5.3 GENERALIZATION ABILITY

**Cross-Species Transfer** To assess the generalization ability of our proposed DDL approach beyond rice, we conducted cross-species experiments on Wheat and Maize. For Wheat we further collected a high-accuracy dataset of 500 samples and a low-accuracy dataset of 52,450 samples; for Maize the high-accuracy dataset contains 567 samples and the low-accuracy dataset contains 38,492 samples. We retrained and evaluated our full DDL model on each species and compared its predictive performance against a broad set of baselines. Tables 8 and 9 summarize the results for each method. From the results shown in these tables, it's evident that—even when evaluated on other crops—our DDL model remains highly competitive.

**Sparse-Region Generalization** To evaluate model robustness across regions with varying data densities, we designed a validation experiment based on spatial grid partitioning. Specifically, the geographic space was divided into grids with a resolution of $1° \times 1°$ in both latitude and longitude,

Table 3: All Performance Comparison of Various Models on Regression Tasks for Wheat

| Model | MAE ↓ | RMSE ↓ | $R^2$ ↑ | MAPE ↓ | Reference |
|---|---|---|---|---|---|
| **ML Models** | | | | | |
| Gradient Boosting R | 1143.22 | 1576.82 | 0.4907 | 0.2698 | (Friedman, 2001) |
| LightGBM | **1061.73** | 1445.15 | **0.5722** | 0.2651 | (Ke et al., 2017) |
| MHA-MLP | 1200.85 | 1579.87 | 0.4802 | 0.2707 | - |
| **Related Work** | | | | | |
| PBA-MLP | 1134.83 | 1508.56 | 0.4750 | 0.2638 | (Ho et al., 2019) |
| Broad Learning System | 1197.73 | 1613.30 | 0.4580 | 0.2839 | (Liu & Chen, 2018) |
| **DDL Model** | | | | | |
| Ours(Full) | 1118.22 | **1434.27** | 0.5590 | **0.2641** | - |

Table 4: All Performance Comparison of Various Models on Regression Tasks for Maize

| Model | MAE ↓ | RMSE ↓ | $R^2$ ↑ | MAPE ↓ | Reference |
|---|---|---|---|---|---|
| **ML Models** | | | | | |
| Gradient Boosting R | 1513.57 | 2013.99 | **0.6838** | 0.3206 | (Friedman, 2001) |
| LightGBM | 1506.19 | 2032.21 | 0.6781 | 0.3251 | (Ke et al., 2017) |
| Random Forest | 1561.46 | 2063.54 | 0.6681 | 0.3350 | (Breiman, 2001) |
| **Related Work** | | | | | |
| PBA-MLP | 2115.14 | 2706.12 | 0.3155 | 0.6853 | (Ho et al., 2019) |
| Broad Learning System | 1672.19 | 2335.77 | 0.3393 | **0.2178** | (Liu & Chen, 2018) |
| **DDL Model** | | | | | |
| Ours (Full) | **1382.21** | **1847.46** | 0.6735 | 0.2542 | - |

and the density of both high- and low-accuracy data within each grid was quantified. Based on these densities, we constructed different testing scenarios (Table **??**).

As shown in the results, reducing the quantity of high-accuracy data (transitioning from Case A to Case B) results in a moderate performance decrease, with $R^2$ dropping from 0.7480 to 0.7115. While this indicates that high-accuracy data contributes to refining predictions, the model maintains a relatively high performance level. In stark contrast, limiting the availability of low-accuracy data (Case C) leads to a substantial deterioration in performance ($R^2$ plummets to 0.2644), even when high-accuracy data is abundant. These findings refute the notion that high-accuracy data alone is sufficient for broad generalization and underscore that the massive low-accuracy dataset acts as a critical stabilizer, enabling the DDL framework to generalize effectively even in regions where empirical observations are sparse.

Table 5: Generalization Ability of the DDL Framework

| Data Configuration | MAE ↓ | RMSE ↓ | $R^2$ ↑ | MAPE ↓ |
|---|---|---|---|---|
| Case A | 810.27 | 1091.62 | 0.7480 | 0.1308 |
| Case B | 616.35 | 867.91 | 0.7115 | 0.0835 |
| Case C | 1386.74 | 1944.45 | 0.2644 | 0.2649 |
| Case D | – | – | – | – |

**Note:** Grids were ranked by the **density of high-accuracy samples**.
– **Top 50%**: Regions with high density of empirical observations (Data-Rich).
– **Bottom 50%**: Regions with low density of empirical observations (Data-Sparse).
*Configuration Details:*
– **Case A**: Full low-accuracy data + High-accuracy data from Top 50% regions.
– **Case B**: Full low-accuracy data + High-accuracy data from Bottom 50% regions.
– **Case C**: Sparse low-accuracy data + High-accuracy data from Top 50% regions.

## 5.4 SENSITIVITY TO NOISE IN LOW-ACCURACY DATA

To test whether the predictive accuracy of our DDL model is tightly constrained by the quality of the low-accuracy dataset, we injected zero-mean Gaussian noise into the low-accuracy data at relative magnitudes of 0%, 10%, 20%, 30% and 40% (noise standard deviation expressed as a fraction of the original signal standard deviation). We retrained and evaluated the full DDL model under each noise condition and compared its $R^2$ against representative baselines. Results in Table 6 show that even with 20% Gaussian noise the DDL model outperforms the strongest baselines, indicating limited sensitivity to simulation noise in the low-accuracy source.

Table 6: Robustness to Noise Injected into Low-Accuracy Data ($R^2$)

| Model / Condition | $R^2$ ↑ | Reference |
|---|---|---|
| **Machine-Learning Baselines** | | |
| Gradient Boosting Regression | 0.6348 | (Friedman, 2001) |
| LightGBM | 0.6186 | (Ke et al., 2017) |
| XGBoost Regression | 0.5757 | (Chen & Guestrin, 2016) |
| **Related Work** | | |
| UNet-ConvLSTM | 0.5497 | (Kamangir et al., 2025) |
| Broad Learning System | 0.5074 | (Liu & Chen, 2018) |
| **DDL (Ours) — noise levels applied to low-accuracy data** | | |
| Ours (0% noise) | **0.6837** | - |
| Ours (10% noise) | **0.6681** | - |
| Ours (20% noise) | **0.6452** | - |
| Ours (30% noise) | 0.6134 | - |
| Ours (40% noise) | 0.5662 | - |

## 6 CONCLUSION AND FUTURE WORK

This study proposes a Deep Learning framework with Dynamic Dropping mechanism, designed to achieve a deep and organic integration of large-scale low-accuracy datasets with small-scale high-accuracy datasets. The framework provides an effective solution for further improving both the accuracy and generalization ability of crop yield prediction models. At its core, DDL introduces a dynamic dropping strategy, which contrasts with traditional static data-cleaning approaches that discard noisy samples or retain uninformative data prior to training (Ashfaq et al., 2025). By embedding data handling as a continuous, dynamic process throughout training, DDL enhances generalization and stability on high-accuracy test sets. This strategy mitigates the inherent trade-off between data quantity and data quality, reducing the reliance on stringent requirements for high-quality datasets. Consequently, training with heterogeneous data sources becomes more reliable in domains such as agriculture and industry, where access to high-quality data is limited (Paudel et al., 2022).

Despite these promising results, three directions remain for further exploration. Firstly, the dropping strategy in this work primarily relies on absolute prediction error, without accounting for intrinsic data distribution characteristics. Future research may incorporate distributional differences between datasets when estimating dropping probabilities (Egele et al., 2024). Secondly, the current weight adjustment scheme follows a linear schedule. Although it alleviates the need for manual weight tuning and provides some adaptivity, it lacks adjustments based on real-time training dynamics. Closing this loop through validation-based feedback mechanisms would be critical for further improving predictive accuracy (Caljon et al., 2025). Eventually, while this study focuses on rice yield prediction in agriculture, the proposed framework could be extended to other domains, offering a generalizable solution for integrating heterogeneous data and enhancing model generalization (Zhang et al., 2025).

In summary, the DDL framework presents a practical approach to jointly train on small high-accuracy and large low-accuracy datasets; empirical results on crop yield tasks show consistent improvements over examined baselines. While these results are encouraging, we avoid broad claims about universal generalizability and instead emphasize that DDL is a promising mechanism whose applicability to other domains should be explored in future work.

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

# A APPENDIX

## A.1 DATA DISTRIBUTION

The following shows the average yield per mu of the Mixed-accuracy dataset across four rice-growing regions and its distribution across various provinces, as illustrated in Figure 4a.

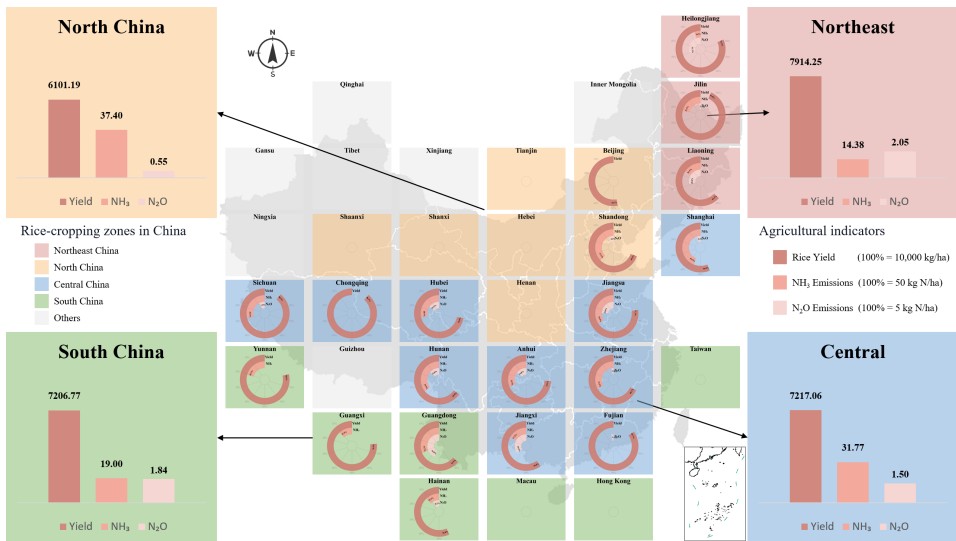

(a) Prediction performance on the Mixed-accuracy dataset (low-accuracy subset).

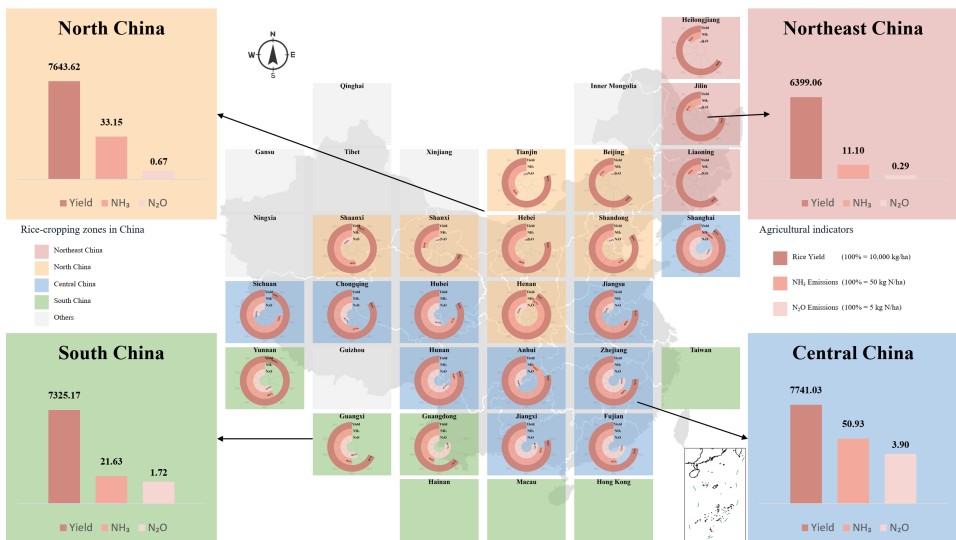

(b) Prediction performance on the Mixed-accuracy dataset (high-accuracy subset).

Figure 4: Comparison of model prediction accuracy across subsets of the Mixed-accuracy dataset.

## A.2 TRAINING DETAILS

To further investigate model behavior, we present two complementary visualizations: one characterizing prediction fidelity across samples, and another revealing the contribution of input features to the model's decisions.

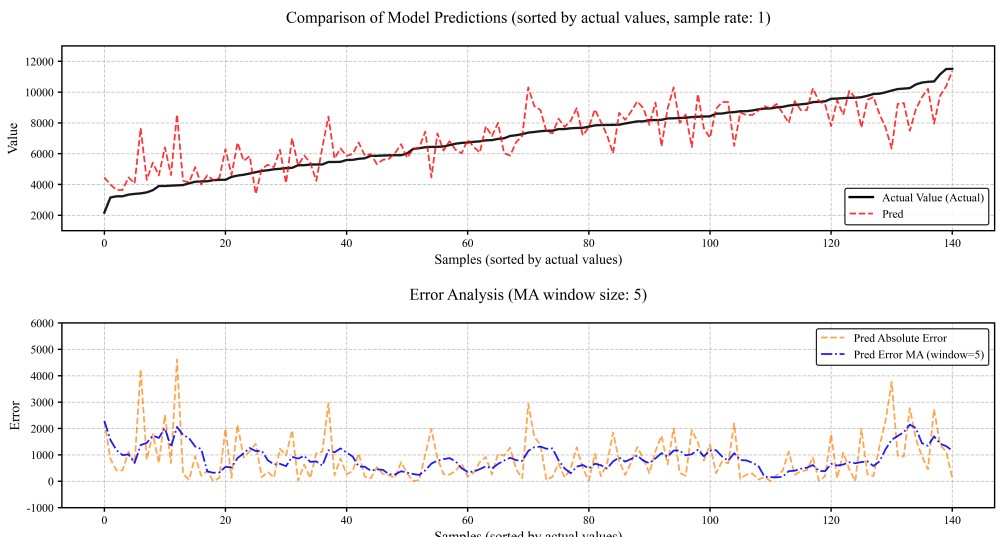

Figure 5: Temporal-style prediction and error analysis across samples. The upper panel displays the actual rice yield (solid black line) alongside model predictions (dashed blue line) ordered by sample index. The lower panel shows the per-sample absolute error (orange dashed line) and its moving average (blue dash-dot line), computed using a sliding window to suppress noise and highlight systematic bias or regional error patterns. A larger window width improves trend visibility but reduces sensitivity to local error spikes.

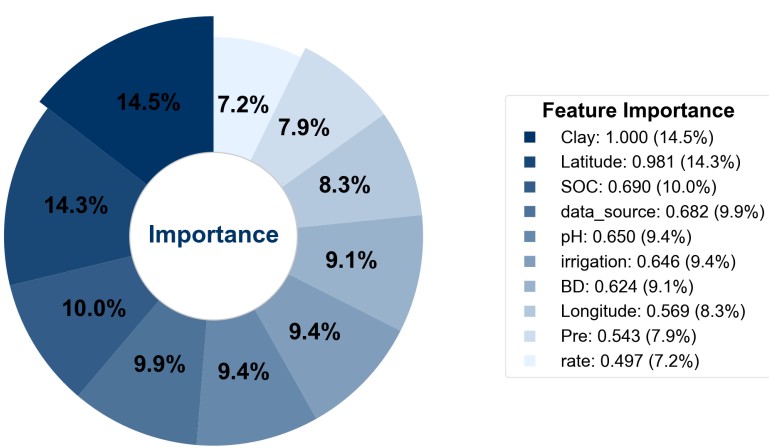

Figure 6: Relative importance of input features in the predictive model, visualized as a donut chart. Features are ranked by their contribution to prediction accuracy, with the top- and bottom-ranked features explicitly labeled and color-highlighted. The hollow center enhances visual focus on the proportional influence of each variable, underscoring which agronomic or environmental factors drive model performance.

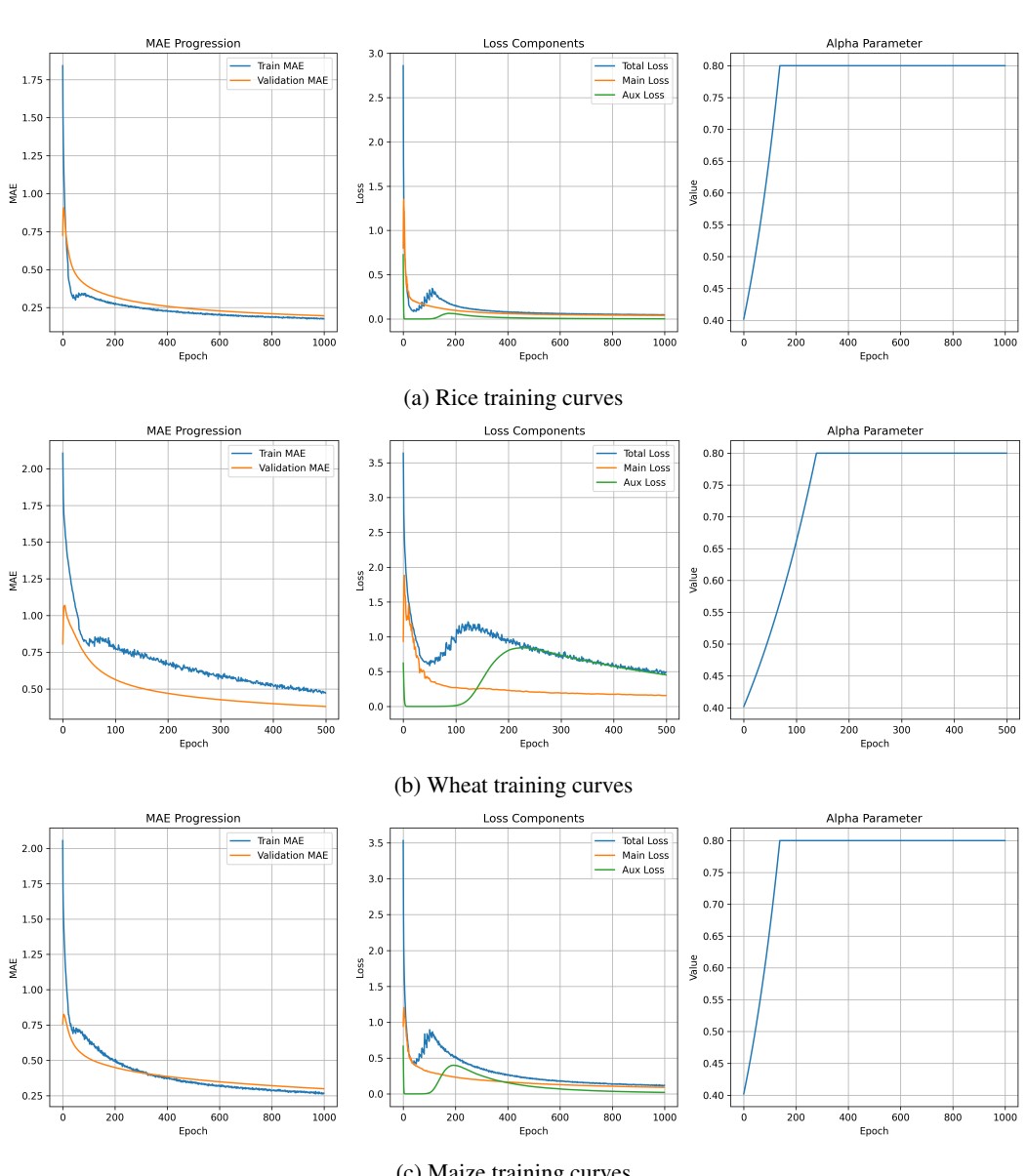

(a) Rice training curves

(b) Wheat training curves

(c) Maize training curves

Figure 7: Training curves of the DDL framework across three major crops. Each subplot shows three key metrics: (1) MAE progression for both training and validation sets, (2) Loss components including total loss, main loss (yield prediction), and auxiliary loss (data source classification), and (3) Alpha parameter evolution that dynamically balances the main and auxiliary tasks during training. The consistent convergence patterns across all three crops validate the generalizability of the DDL framework for different crop yield prediction scenarios.

A.3  RESULT DETAILS

Tables 7, 8, and 9 comprehensively evaluate the predictive performance of our DDL framework against various baseline models across three major crops: rice, wheat, and maize. These tables present detailed comparisons using multiple evaluation metrics. The results validate the generalizability of the DDL framework beyond rice to other important cereal crops.

Table 7: All Performance Comparison of Various Models on Regression Tasks for Rice

| Model | MAE ↓ | RMSE ↓ | $R^2$ ↑ | MAPE ↓ | Reference |
|---|---|---|---|---|---|
| **ML Model** | | | | | |
| Decision Tree R | 1098.62 | 1495.59 | 0.5076 | 0.1752 | (Breiman et al., 1984) |
| Gradient Boosting R | 1001.96 | 1287.90 | 0.6348 | 0.1648 | (Friedman, 2001) |
| LightGBM | 1011.40 | 1316.29 | 0.6186 | 0.1678 | (Ke et al., 2017) |
| Linear Regression | 1344.19 | 1684.27 | 0.3755 | 0.2309 | (Seber & Lee, 2012) |
| Random Forest | 1076.34 | 1436.98 | 0.5454 | 0.1754 | (Breiman, 2001) |
| SVR | 1777.09 | 2123.63 | 0.0072 | 0.3108 | (Drucker et al., 1996) |
| XGBoost Regression | 1064.40 | 1388.34 | 0.5757 | 0.1750 | (Chen & Guestrin, 2016) |
| Deep Learning Regression | 2938.72 | 3500.50 | -1.6976 | 0.4183 | (Goodfellow et al., 2016) |
| GPR | 1296.37 | 1808.19 | 0.3381 | 0.2098 | (Rasmussen & Williams, 2006) |
| MHA-MLP | 1115.07 | 1496.70 | 0.5465 | 0.1767 | - |
| **Related Work** | | | | | |
| CNN+GAN | 1115.56 | 1637.84 | 0.3587 | 0.2026 | (Zhang et al., 2022b) |
| UNet-ConvLSTM | 1075.91 | 1357.26 | 0.5497 | 0.1672 | (Kamangir et al., 2025) |
| Remote Sensing + SCE-UA | 4025.99 | 5391.50 | -2.7046 | 0.7144 | (Li et al., 2024) |
| PBA-ResNet | 1082.35 | 1462.31 | 0.4823 | 0.1612 | (He et al., 2016) |
| PBA-MLP | 1108.91 | 1642.98 | 0.4535 | 0.1635 | (Ho et al., 2019) |
| Broad Learning System | 1160.32 | 1559.85 | 0.5074 | 0.1815 | (Liu & Chen, 2018) |
| **Process Model** | | | | | |
| DNDC Model | 1439.35 | 1754.55 | 0.2910 | 8.7361 | (Li et al., 1992) |
| **DDL Model** | | | | | |
| Ours(High-Accuracy Only) | 958.62 | 1335.92 | 0.6158 | 0.1602 | - |
| Ours(w/o Dynamic Discard) | 887.10 | 1273.89 | 0.6506 | 0.1477 | - |
| Ours(w/ Static Weighting) | 870.31 | 1249.90 | 0.6637 | 0.1439 | - |
| Ours(Full) | **852.30** | **1212.14** | **0.6837** | **0.1424** | - |

Table 8: All Performance Comparison of Various Models on Regression Tasks for Wheat

| Model | MAE ↓ | RMSE ↓ | $R^2$ ↑ | MAPE ↓ | Reference |
|---|---|---|---|---|---|
| **ML Model** | | | | | |
| Decision Tree R | 1326.32 | 1929.20 | 0.2377 | 0.3192 | (Breiman et al., 1984) |
| Gradient Boosting R | 1143.22 | 1576.82 | 0.4907 | 0.2698 | (Friedman, 2001) |
| LightGBM | **1061.73** | 1445.15 | **0.5722** | 0.2651 | (Ke et al., 2017) |
| Linear Regression | 1454.95 | 1817.08 | 0.3237 | 0.3653 | (Seber & Lee, 2012) |
| Random Forest | 1143.93 | 1617.93 | 0.4638 | 0.2849 | (Breiman, 2001) |
| SVR | 1819.56 | 2213.53 | -0.0036 | 0.5500 | (Drucker et al., 1996) |
| XGBoost Regression | 1129.97 | 1634.62 | 0.4527 | 0.2770 | (Chen & Guestrin, 2016) |
| Deep Learning Regression | 2967.98 | 3651.12 | -1.7305 | 0.5131 | (Goodfellow et al., 2016) |
| GPR | 1557.16 | 1945.91 | 0.2115 | 0.4009 | (Rasmussen & Williams, 2006) |
| MHA-MLP | 1200.85 | 1579.87 | 0.4802 | 0.2707 | - |
| **Related Work** | | | | | |
| PBA-MLP | 1134.83 | 1508.56 | 0.4750 | 0.2638 | (Ho et al., 2019) |
| Broad Learning System | 1197.73 | 1613.30 | 0.4580 | 0.2839 | (Liu & Chen, 2018) |
| **DDL Model** | | | | | |
| Ours(Full) | 1118.22 | **1434.27** | 0.5590 | **0.2641** | - |

Table 9: All Performance Comparison of Various Models on Regression Tasks for Maize

| Model | MAE ↓ | RMSE ↓ | $R^2$ ↑ | MAPE ↓ | Reference |
|---|---|---|---|---|---|
| **ML Model** | | | | | |
| Decision Tree R | 1936.85 | 2685.83 | 0.4377 | 0.3633 | (Breiman et al., 1984) |
| Gradient Boosting R | 1513.57 | 2013.99 | **0.6838** | 0.3206 | (Friedman, 2001) |
| LightGBM | 1506.19 | 2032.21 | 0.6781 | 0.3251 | (Ke et al., 2017) |
| Linear Regression | 2215.98 | 2901.25 | 0.3439 | 0.4816 | (Seber & Lee, 2012) |
| Random Forest | 1561.46 | 2063.54 | 0.6681 | 0.3350 | (Breiman, 2001) |
| SVR | 2656.28 | 3589.16 | -0.0042 | 0.6257 | (Drucker et al., 1996) |
| XGBoost Regression | 1635.27 | 2188.98 | 0.6265 | 0.3221 | (Chen & Guestrin, 2016) |
| Deep Learning Regression | 6155.25 | 6969.60 | -2.7866 | 0.7119 | (Goodfellow et al., 2016) |
| GPR | 1901.65 | 2587.13 | 0.1895 | 0.2673 | (Rasmussen & Williams, 2006) |
| MHA-MLP | 1841.45 | 2493.63 | 0.2470 | 0.2619 | - |
| **Related Work** | | | | | |
| PBA-MLP | 2115.14 | 2706.12 | 0.3155 | 0.6853 | (Ho et al., 2019) |
| Broad Learning System | 1672.19 | 2335.77 | 0.3393 | **0.2178** | (Liu & Chen, 2018) |
| **DDL Model** | | | | | |
| Ours(Full) | **1382.21** | **1847.46** | 0.6735 | 0.2542 | - |

## B. ETHICS STATEMENT

This research adheres to the ICLR Code of Ethics. No human subjects or animal experiments were involved in this study. All datasets used were obtained in accordance with their respective usage guidelines to ensure no violation of privacy. We have made every effort to avoid bias or discriminatory outcomes in our research. No personally identifiable information was used, and no experiments were conducted that could raise privacy or security concerns. We are committed to maintaining transparency and integrity throughout this research.

## C. REPRODUCIBILITY STATEMENT

We have made every effort to ensure that the results presented in this paper are reproducible. All code and datasets have been submitted as supplementary materials to facilitate replication and verification by others.

## D. LARGE LANGUAGE MODEL (LLM) USAGE STATEMENT

A large language model (LLM) was used to assist in the writing and editing of this manuscript. Specifically, we employed an LLM to help improve language expression, enhance readability, and ensure clarity across all sections of the paper. The model provided support in tasks such as sentence rephrasing, grammar checking, and improving overall textual fluency.

It is important to emphasize that the LLM did not contribute to the conception of research ideas, methodology, or experimental design. All research concepts, insights, and analyses were independently developed and conducted by the authors. The LLM's role was strictly limited to enhancing the linguistic quality of the manuscript and did not involve any scientific content or data analysis.

The authors take full responsibility for all content in the manuscript, including any text generated or refined by the LLM. We have ensured that all LLM-assisted text complies with ethical standards and does not constitute plagiarism or academic misconduct.

