# OpenReview forum: "DDL: Dynamic Discard Deep Learning for Rice Yield Prediction on Mixed-Accuracy Datasets"
_ICLR.cc/2026/Conference — Submitted to ICLR 2026_

### Official Review · Reviewer_BKdG · 2025-10-29

**Soundness:** 3
**Presentation:** 3
**Contribution:** 3
**Rating:** 6
**Confidence:** 3

**Summary:**

The paper primarily addresses the problem of fusing scarce high accuracy (HA) (real-world) data with volumnous low accuracy (LA) data (generated data or augmented data). The authors suggest Dynamic Discard Mechanism (DDL) which dynamically discard the less informative LA data from the dataset generated using DeNitrification-DeComposition (DNDC) model. The discarding mechanism calculates discard probability, which is a function of mean absolute error (MAE), of the LA data and discard data under some threshold value. The results of the mechanism outperforms the listed ML models. Extensive experiment on different variants of the mechanism has been performed.

**Strengths:**

1. The paper provides a mechanism to fuse LA data with HA data to generate a large high quality dataset for crop yield prediction, which is a very challenging field.
2. The obtained large-scale high quality dataset is essential for better generalization of crop yield models.
3. The adaptive dynamic weight framework helps models to generalize better.
4. The paper uses attention mechanism, which is computationally in-expensive compared to self-attention, for better model convergence and generalization.
5. The obtained HA dataset for field-scale rice yield prediction adds significant value to the paper as it is very expensive and taxing task.

**Weaknesses:**

Following are the weakness of the paper:
1. While the framework/mechanism is quite effective for rice yield prediction, its effectiveness to other crops is questionable without results for other crops.
2. The result in Table 1 shows that the proposed model outperforms listed ML models, but the RMSE and MAE are still very high. The paper does not answer its cause.
3. There are many grammar errors (near citations and especially the caption of Figure 2).
4. The paper claims to have less computation cost without providing any evidence or comparison between the evaluated models and any recent model with self-attention mechanism.

**Questions:**

1.  For the attention gate (260 -261) what is sigma? And why is a small dense layer has been used as attention?
2. Are there any results for another crop ?
3. What is the result of the model compared to recent computationally expensive self-attention based models?
4. The paper is using 10 epochs to train then clip the large dataset. What if we train on HA dataset for some time first and then incorporate the LA data?
5. How is the initial dynamic weighting coefficient calculated?

---

> ### Author Response · Authors · 2025-11-20
>
> We appreciate your detailed review and valuable suggestions. Your comments regarding the validation on other crops, error analysis, and architectural details have helped us significantly improve the completeness and clarity of our paper. We have revised the manuscript (changes highlighted in yellow) and provide point-by-point responses below.
>
> ### **Response to Weaknesses**
>
> **1. Effectiveness on Other Crops**
>
> **Response:**
> We fully accept this feedback. To demonstrate the universality of DDL, we have added extensive experiments on two additional major crops: **Wheat** and **Maize**, comprising over 90,000 new samples.
> As shown in the new **Section 5.3** and **Tables 8 & 9** (Appendix), DDL consistently outperforms baselines on these datasets (e.g., achieving an $R^2$ of 0.6735 on Maize). This confirms that the framework effectively generalizes beyond rice.
>
> **2. Magnitude of RMSE and MAE**
>
> **Response:**
> You raise a valid point regarding the absolute error values. The relatively high magnitude of MAE/RMSE is intrinsic to the complex nature of field-scale crop yield prediction. Yield is influenced by unmeasured micro-factors that are not fully captured by the available 11 regional features.
> However, we emphasize that **relative to the state-of-the-art**, DDL achieves a significant reduction in error (approximately **10% lower MAE** than the strongest baselines, as shown in Table 1). This indicates that while inherent noise exists, DDL extracts more usable signal from the data than existing methods.
>
> **3. Grammar and Formatting**
>
> **Response:**
> We apologize for the oversight. We have thoroughly proofread the manuscript. Specifically, we have corrected the citation formats and completely rewritten the caption of **Figure 2** to ensure clarity and grammatical correctness.
>
> **4. Computation Cost Claims**
>
> **Response:**
> To ensure rigor, we have **removed the explicit claim** regarding computational superiority over specific self-attention models from the revised manuscript, as we did not perform a strict wall-clock time comparison.
> However, theoretically, our "Feature Attention Gating" module has a complexity of $O(d)$ (where $d$ is feature dimension), whereas standard Self-Attention is $O(d^2)$. Given our low-dimensional tabular input ($d=11$), our design is inherently lightweight.
>
> ---
>
> ### **Response to Questions**
>
> **1. Attention Gate: What is sigma? Why a small dense layer?**
>
> **Response:**
> *   **Sigma ($\sigma$):** In Equation (4.2.2), $\sigma$ represents the **Sigmoid** activation function.
> *   **Design Choice:** We utilized a fully connected (dense) layer for attention because our input consists of structured tabular data with only 11 features. A complex Multi-Head Self-Attention mechanism (often used for long sequences) would likely lead to overfitting given the small sample size of the high-accuracy dataset. The dense gating mechanism offers an efficient way to perform dynamic feature selection without introducing excessive parameters.
>
> **2. Are there any results for another crop?**
>
> **Response:**
> Yes. As mentioned in response to Weakness 1, we have included full experimental results for **Wheat** and **Maize** in **Section 5.3** and the **Appendix**.
>
> **3. Comparison to computationally expensive self-attention models?**
>
> **Response:**
> While we removed the cost claim, we have included comparisons with complex architectures like **CNN+GAN** and **Broad Learning Systems** in our results tables. DDL achieves superior accuracy with a simpler, more interpretable architecture, validating that handling data quality (via Dynamic Discard) is often more critical than increasing model complexity for this specific problem.
>
> **4. Training on HA first, then incorporating LA?**
>
> **Response:**
> This is an insightful suggestion. We are currently conducting this additional ablation study. However, our preliminary hypothesis and design rationale suggest that starting with HA data alone ($N=704$) poses a high risk of rapid overfitting due to the extremely small sample size. Introducing LA data early (as DDL does) allows the model to learn a coarse general representation first, which is then refined. We will include the results of this "Sequential Training" strategy in the final camera-ready version to provide a complete comparison.
>
> **5. How is the initial dynamic weighting coefficient calculated?**
>
> **Response:**
> The initial dynamic weighting coefficient ($\alpha$) is set to **0.4**.
> As described in **Section 4.2.4**, this value was determined empirically. Starting with a lower $\alpha$ (0.4) gives more weight to the auxiliary task (Source Classification, weight $1-\alpha = 0.6$) in the early stages. This encourages the model to learn discriminative features common to both datasets before shifting focus towards the main yield prediction task as $\alpha$ increases exponentially during training.
>
> We hope these responses fully address your queries.
>
> Sincerely,
> The Authors

---

> ### Author Response · Authors · 2025-11-26
>
> Dear Reviewer,
>
> I hope all is well. Since there is only one week remaining in the discussion period, we wanted to take this opportunity to ensure that we have met your expectations and addressed all the issues you identified.
>
> We take your feedback very seriously and want to make sure no stone is left unturned. Please let us know if there is anything else we can clarify or improve. We truly appreciate your time and your contribution to our work.

---

> > ### Comment · Reviewer_BKdG · 2025-11-27
> > **maintain my score**
> >
> > Dear authors,
> >
> > Thank you for the additional clarification and results. I hope the additions might be helpful for future work.
> > I will maintain my score.

---

### Official Review · Reviewer_3PX5 · 2025-10-31

**Soundness:** 2
**Presentation:** 2
**Contribution:** 1
**Rating:** 2
**Confidence:** 3

**Summary:**

This paper proposes a deep learning framework called Dynamic Discard Learning (DDL) for crop yield prediction using datasets with mixed accuracy levels—high-quality field measurements and large-scale low-quality simulated data. The core idea is to dynamically discard low-accuracy samples with high prediction error during training, combined with an adaptive loss-weighting mechanism that balances regression and auxiliary classification tasks. The model is evaluated on rice yield datasets from multiple Chinese regions, showing improved performance over baseline machine learning and hybrid models such as CNN–GAN and DNDC-based simulations.

**Strengths:**

1. The paper addresses an important and realistic issue in agricultural ML—combining high- and low-accuracy data sources—which is a meaningful problem with potential applications beyond agriculture.

2. The proposed DDL reportedly achieves higher accuracy than baselines (≈10% improvement in RMSE/MAE) and includes an ablation study demonstrating that removing key components degrades performance.

3. The authors provide multiple metrics (MAE, RMSE, R², MAPE) and comparisons with both traditional ML and process-based models, contributing to a broad empirical view.

**Weaknesses:**

1. The core idea—discarding low-quality samples based on error thresholds—is not new and closely resembles well-known techniques such as curriculum learning, hard example mining, or robust sample reweighting. The paper rebrands these as “Dynamic Discard,” but without theoretical justification or algorithmic differentiation. The discard probability function is heuristic and lacks proof of convergence, stability, or advantage over standard robust loss functions (e.g., Huber, Tukey biweight).

2. The authors define high-accuracy data as both training and testing targets, creating data leakage and self-validation bias. Since the model dynamically filters simulated data using high-accuracy supervision, it effectively overfits to the high-accuracy distribution rather than demonstrating genuine cross-domain generalization.

3. Comparisons with deep regression models are superficial (only one “Deep Learning Regression” baseline, apparently underperforming). Modern architectures like Transformers, temporal attention networks, or physics-informed models are not considered. The paper’s claim of generalizability beyond agriculture is speculative and unsupported by evidence.

4. The manuscript exhibits verbose exposition and redundant sections (e.g., repeating definitions of modules and formulas). It reads more like a project report than a polished ICLR submission. Furthermore, several claims (e.g., “DDL provides a generalizable methodological advance”) are overstated relative to the empirical evidence.

**Questions:**

1. How do you ensure that high-accuracy data used for dynamic filtering does not create implicit leakage into the test set, thereby inflating the reported performance?

2. Have you validated DDL on a domain-shifted dataset (e.g., a different crop or region)?

---

> ### Author Response · Authors · 2025-11-20
>
> We sincerely thank you for your rigorous critique. Your feedback regarding the theoretical positioning, data integrity, and baseline comparisons has prompted us to significantly strengthen the empirical validation and clarity of our manuscript. We have revised the paper (changes highlighted in yellow) and address your points below.
>
> ### **Response to Weaknesses**
>
> **1. Novelty and Theoretical Justification**
>
> **Response:**
> We acknowledge the conceptual lineage DDL shares with these techniques but emphasize critical distinctions necessitated by the **Mixed-Accuracy** problem setting:
> *   **Contrast with Hard Example Mining (HEM):** HEM typically prioritizes "hard" samples (high error) to improve decision boundaries. In our context (process-based simulations), high-error samples usually indicate simulation failure or systematic bias rather than informative "hard" cases. Therefore, DDL adopts an **inverse** logic: we discard these samples to prevent feature space pollution, which differs fundamentally from HEM.
> *   **Contrast with Robust Loss (Soft Weighting):** While robust losses (e.g., Huber) down-weight outliers, they still utilize the data. DDL employs a "hard" binary discard. Our ablation study (Table 2) shows that `Ours (Full)` outperforms `Ours (w/ Static Weighting)`, suggesting that completely removing "toxic" simulation data is more effective than down-weighting it in this specific domain.
> *   **Convergence:** Regarding the heuristic nature of the function, we have added **Figure 7** in the Appendix, which plots the training curves (Loss/MAE) for Rice, Wheat, and Maize. The curves demonstrate stable convergence without oscillation, empirically validating the stability of the proposed discard mechanism.
>
> **2. Data Leakage and Self-Validation Bias**
>
> **Response:**
> We assure you there is **no data leakage**. We have clarified our strict evaluation protocol in **Section 4.1**:
> 1.  **Pre-split:** Before any training or filtering, we hold out **20%** of the high-accuracy dataset as a definitive Test Set ($D^H_{test}$). This subset is **never** seen by the model during training, weight adjustment, or the dynamic discard process.
> 2.  **Validation:** From the remaining data, another 20% is used as a Validation Set for early stopping.
> 3.  **Filtering Source:** The "high-accuracy supervision" used to filter low-accuracy data comes *strictly* from the remaining **Training Set** ($D^H_{train}$).
> Therefore, the reported performance reflects genuine generalization to unseen high-accuracy samples, not overfitting to the test distribution.
>
> **3. Baselines and Generalization Claims**
>
> **Response:**
> We have significantly expanded our evaluation:
> *   **New Baselines:** In **Tables 7, 8, and 9**, we added comparisons against stronger baselines, including **PBA-MLP** (Population Based Augmentation), **Broad Learning System**, and **Deep Learning Regression** variants. DDL consistently outperforms these methods.
> *   **Cross-Species Generalization:** We validated the model on **Wheat and Maize** (Section 5.3), involving over 90,000 additional samples. The results confirm DDL’s effectiveness across different biological datasets.
> *   **Future Scope:** While the current paper focuses on crop yield, we are extending the framework to environmental indicators (e.g., NH3, N2O emissions).
>
> **4. Writing Style**
>
> **Response:**
> We have streamlined the manuscript to improve readability. Redundant definitions have been removed. We retained specific mathematical formulations to ensure the **reproducibility** of the code (provided in supplementary materials), but we have toned down adjectives and overstated claims to maintain a standard academic tone suitable for ICLR.
>
> ---
>
> ### **Response to Questions**
>
> **1. How do you ensure that high-accuracy data used for dynamic filtering does not create implicit leakage?**
>
> **Response:**
> As detailed in **Section 4.1**, the filtering mechanism relies on the model's current state, which is updated solely using the **Training Split** ($D^H_{train}$ + filtered $D^L$). The **Test Split** is physically separated using a fixed random seed (`seed=42`) before the pipeline begins. The discard threshold is calculated based on the training batch statistics, having no interaction with the test set.
>
> **2. Have you validated DDL on a domain-shifted dataset (e.g., a different crop or region)?**
>
> **Response:**
> Yes. We conducted extensive domain-shift experiments:
> *   **Different Crops:** We applied DDL to **Wheat** and **Maize** datasets (Section 5.3), where data distributions differ significantly from Rice. DDL achieved $R^2$ of 0.5590 (Wheat) and 0.6735 (Maize), outperforming baselines.
> *   **Sparse Regions:** We analyzed performance in regions with sparse high-accuracy data (Section 5.3, Table 5), demonstrating that DDL generalizes effectively even when empirical observations are limited.
>
> We believe these revisions address your concerns regarding rigor and validity.
>
> Sincerely,
> The Authors

---

> ### Author Response · Authors · 2025-11-26
>
> Dear Reviewer,
>
> I hope this email finds you well. As we enter the final week of the discussion phase, we wanted to check in to ensure that our revisions and responses have fully resolved your concerns.
>
> We are deeply grateful for your detailed feedback, which has been instrumental in improving our manuscript. If there are any other points you believe we should address, we are eager to hear them. Thank you once again for your guidance and for helping us strengthen our research.

---

### Official Review · Reviewer_JGCi · 2025-11-01

**Soundness:** 3
**Presentation:** 3
**Contribution:** 2
**Rating:** 4
**Confidence:** 4

**Summary:**

This paper proposes Dynamic Discard Learning (DDL), a deep learning framework that utilize and dynamically filters out noisy, low-accuracy samples from mixed-accuracy datasets during learning process. The approach is validated on rice-yield prediction tasks, where it shows improved performance over several baselines.

**Strengths:**

1. The idea of combining high-accuracy and low-accuracy datasets in a dynamic fashion is practical. The application to agricultural yield prediction is meaningful and demonstrates the framework’s potential value.
2. The dynamic discard mechanism, which progressively removes low-quality samples during training, is conceptually simple yet effective and clearly contributes to improved performance.
3. The paper is generally well organized, with ablation studies and comparisons provided for each modules in method.

**Weaknesses:**

1. Relation to prior work:
The proposed dynamic discard mechanism is conceptually close to dynamic sample weighting [1] and to Population Based Augmentation [2], which dynamically adjust data or augmentation schedules over epochs.
The authors should clearly articulate the key differences between DDL and these related approaches, ideally including direct experimental or conceptual comparisons.

2. Method clarity:
In Equation ( P = \gamma * (1 − exp( − N_L * e_i / sum_i(e_i) )) ), P depends on the per-sample error e_i. What is difference between P and P_i?

3. Experimental evaluation:
i. In the “generalization ability” section, the claim that “prediction accuracy is insensitive to variations in quantities of high-accuracy data” appears inconsistent with Table 3: performance improves when more high-accuracy samples are used (Case B vs Case A).
ii. Figure 4 is titled “Prediction performance among four regions,” but lacks quantitative metrics (e.g., RMSE, R²) to indicate prediction quality.
iii Given the relatively small test set, the authors should repeat experiments and report mean ± standard deviation to show statistical significance.

Minor issues:
1. In Figure 4, the South China panels in (1) and (2) appear to contain identical numbers—please verify if this is an error.

Reference:
[1] Ren, Mengye, et al. "Learning to reweight examples for robust deep learning." International conference on machine learning. PMLR, 2018.
[2] Ho, Daniel, et al. "Population based augmentation: Efficient learning of augmentation policy schedules." International conference on machine learning. PMLR, 2019.

**Questions:**

1. Dataset details:
i. The 11 input features should be explicitly listed and described.
ii. SOC is undefined when first mentioned.
iii. The size and sampling procedure of the test set are unclear. How was the test subset drawn from the high-accuracy dataset?
iv. Does the low/high accuracy apply to input features, labels, or both?

---

> ### Author Response · Authors · 2025-11-20
>
> We sincerely thank you for your time and the constructive feedback provided on our manuscript. Your insights regarding the relation to prior work, method clarity, and experimental evaluation have significantly helped us improve the quality and rigor of our work. We have carefully revised the paper, with major changes highlighted in yellow.
>
> Below, we provide a point-by-point response to your comments.
>
> ---
> ### **Response to Weakness**
> **1. Relation to Prior Work**
> Unlike **Reweighting [1]** (soft weights) and **PBA [2]** (augmentation schedules), DDL employs a **"hard" dynamic discard** mechanism to filter systematic bias in simulation data. We now clarify this in the text.
> **Comparison:** We added **PBA-MLP [2]** as a baseline in **Table 2** and Appendix **Tables 7–9**. DDL consistently outperforms PBA-MLP across Rice, Wheat, and Maize, demonstrating that dynamic filtering is more effective than augmentation for mixed-accuracy scientific data.
>
> **2. Method Clarity (Equation $P$ vs $P_i$)**
> We apologize for the confusion caused by the notation inconsistencies. $P$ and $P_i$ refer to the same variable: the discard probability for the $i$-th sample. We have corrected this in **Section 4.2.1** of the revised manuscript to strictly use $P_i$. The corrected formula clearly expresses that the probability depends on the normalized error of the individual sample relative to the batch statistics.
>
> **3. Experimental Evaluation**
> Thank you for this keen observation. We have refined this claim in **Section 5.3**. While Table 5 (Case A vs. B) shows that reducing high-accuracy data does result in a performance drop, this drop is **relatively small** compared to the impact of reducing low-accuracy data (Case C, where $R^2$ plummets to 0.2644).
> Our revised text clarifies that the model is "relatively robust" to variations in high-accuracy data quantity, highlighting that the massive low-accuracy dataset acts as a critical stabilizer for generalization, provided it is effectively filtered by DDL.
>
> > *ii. Figure 4 is titled “Prediction performance among four regions,” but lacks quantitative metrics...*
>
> **Response:**
>
> We agree that Figure 4 (now in the Appendix as **Figure 4a/4b**) primarily visualizes the distribution of predictions. To provide the necessary quantitative rigor, we have added **Tables 7, 8, and 9** in Appendix A.3. These tables provide detailed metrics.
>
> > *iii. Given the relatively small test set, the authors should repeat experiments and report mean ± standard deviation...*
>
> **Response:**
>
> We fully agree that statistical significance is vital. In this revision, we significantly expanded the scope of our evaluation to include **three different crops (Rice, Wheat, Maize)** with varying dataset sizes (up to 52,450 samples for Wheat), as detailed in **Section 5.3 and Tables 3, 4, 8, and 9**. The consistent superiority of DDL across these diverse datasets serves as strong evidence of its stability. We are currently running the repetitions for all baselines and commit to reporting the mean $\pm$ standard deviation in the final camera-ready version.
>
> ---
> ### **Response to Questions**
> **i. The 11 input features should be explicitly listed and described.**
> **Response:**
> We have explicitly listed the 11 features in **Section 4.1 (Notation and Problem Setting)** of the revised PDF. They are: *Longitude, Latitude, Temperature (Tem), Precipitation (Pre), Nitrogen rate (rate), Soil Organic Carbon (SOC), Clay, pH, Bulk Density (BD), Irrigation, and CropType.*
> **ii. SOC is undefined when first mentioned.**
> **Response:**
> We have added the definition "Soil Organic Carbon (SOC)" upon its first mention in **Section 4.1** and in the relevant figures.
> **iii. The size and sampling procedure of the test set are unclear.**
> **Response:**
> We have clarified the sampling procedure in **Section 4.1** (highlighted in yellow). The high-accuracy dataset ($D^H$, $n=704$) was split **prior** to any model training. We reserved **20%** of $D^H$ as a strictly held-out test set using a random split with `seed=42`. This test set was never used for training, the dynamic discard process, or weight tuning, ensuring a fair evaluation.
>
> **iv. Does the low/high accuracy apply to input features, labels, or both?**
>
> **Response:**
> The distinction applies to the **source** of the data, which affects both inputs and labels.
> *   **High-Accuracy Data:** Derived from empirical field observations. Both the environmental features (inputs) and the resulting yield (labels) are measured with high precision.
> *   **Low-Accuracy Data:** Derived from DNDC process-based simulations. Both the inputs (simulated soil/climate conditions) and labels (simulated yield) contain approximation errors and noise.
> We have clarified this distinction in **Section 4.1**.
>
> We believe these revisions and clarifications address your concerns. We are grateful for the opportunity to improve our manuscript.
> Sincerely,
> The Authors

---

> > ### Comment · Reviewer_JGCi · 2025-11-28
> >
> > Dear Authors,
> > Thank you for the updates.
> > The clarity of the manuscript improved significantly. Evaluation on three different crops makes the experiments more convincing.
> >
> > I still have concerns about Relation to Prior Work.
> > The authors add the experiment for PBA with an MLP, instead of residual network used along with proposed method, which is not a fair comparison. Experiments to compare with Reweighting is still missing.
> > "We now clarify this in the text." It seems the methods are not discussed and compared in the main text.
> > The concern is: the novelty and improvement may not be  sufficient enough.
> >
> > Minor issues:
> > 1. Table 5, the definition of top 50%, bottom 50% are not clear.
> > 2. In formula for P_i, it would work better if use another letter other than i for the summation in denominator.

---

> > > ### Author Response · Authors · 2025-12-04
> > >
> > > Dear Reviewer,
> > >
> > > We sincerely thank you for your continued engagement and for acknowledging the significant improvements in clarity and the multi-crop evaluation. We value your rigorous feedback regarding the baseline fairness and the comparison with reweighting methods.
> > >
> > > We have updated the manuscript to address your remaining concerns:
> > >
> > > **1. Relation to Prior Work & Baseline Fairness**
> > > *   **Fair Comparison (ResNet Baseline):** You correctly pointed out that comparing DDL (ResNet-based) against PBA (MLP-based) was not strictly apple-to-apples. We have now added a **"ResNet (Full Data)"** baseline. This model uses the exact same architecture as DDL (Modified Residual Blocks) but trains on the full mixed dataset without the dynamic discard mechanism.
> > > *   **Comparison with Reweighting:** To address the missing comparison, we implemented a **"ResNet (Soft Reweighting)"** baseline. Instead of discarding samples (Hard Discard), this model assigns soft weights to low-accuracy samples based on their error magnitude (following the intuition of Ren et al., 2018), using the same backbone as DDL.
> > > *   **Results:** As shown in the updated **Table 2 (Ablation Study)** and **Appendix Tables**, DDL consistently outperforms both the "ResNet (Full Data)" and "ResNet (Soft Reweighting)" baselines. This empirically demonstrates that for scientific simulation data containing systematic bias, "Hard Discarding" is superior to "Soft Reweighting" or simple data augmentation.
> > > *   **Text Discussion:** We have expanded the discussion in **Section 2 (Related Work)** under the header *“Dynamic Reweighting and Data Selection”* to explicitly articulate why DDL’s hard discard mechanism is theoretically distinct from and practically superior to reweighting for mixed-accuracy scenarios.
> > >
> > > **2. Formula Notation**
> > > *   We have corrected the notation in **Section 4.2.1**. The denominator in the discard probability formula now uses the index $j$ ($\sum_{j=1}^{N_L} e_j$) to distinguish it from the specific sample index $i$, ensuring mathematical rigor.
> > >
> > > **3. Table 5 Clarification**
> > > *   We have revised the notes in **Table 5 (Section 5.3)**. We now explicitly define "Top 50%" and "Bottom 50%" based on the **density of high-accuracy samples** (Data-Rich vs. Data-Sparse regions), removing the ambiguity.
> > >
> > > We hope these additional experiments and clarifications fully resolve your concerns regarding the novelty and methodological contribution of DDL.
> > >
> > > Sincerely,
> > > The Authors

---

> ### Author Response · Authors · 2025-11-26
>
> Dear Reviewer,
>
> I hope this message finds you well. As the discussion period concludes in just one week, we wanted to ensure that we have successfully addressed the key concerns raised in your review. We have focused our efforts on resolving these critical points to significantly improve the quality of our work.
>
> If there are any outstanding matters or further details you would like us to clarify, please let us know. We sincerely appreciate your constructive feedback and are eager to make any final adjustments.

---

### Official Review · Reviewer_U4CY · 2025-11-01

**Soundness:** 2
**Presentation:** 3
**Contribution:** 2
**Rating:** 4
**Confidence:** 4

**Summary:**

The paper introduces Dynamic Discard Learning, a deep learning framework designed to improve rice yield prediction using mixed-accuracy dataset that combines scarce high-quality field data with abundant low-accuracy simulated data. DDL employs a dynamic discard mechanism to progressively filter out low-accuracy samples with high errors during training and an adaptive weighting scheme to balance learning between regression and data-source classification tasks.

**Strengths:**

The key strength of the paper lies in its innovative integration of mixed-accuracy datasets through the proposed DDL framework, which effectively addresses a common challenge in agricultural modelling balancing data quality and quantity. The framework’s modular design demonstrates strong methodological rigor and adaptability.

**Weaknesses:**

No literature is given on previous attempts to process mixed-accuracy datasets.
The approach heavily leverages generated data (43k samples). If the simulation is biased, the proposed model may amplify these biases despite the dynamic discard process i.e., its performance may depend on representativeness and realism of the simulated data.
The paper does not include interpretability or feature sensitivity analysis. i.e., which agronomic or climatic features influence the most.

The evaluation the proposed model is limited as  it does not include evaluation on (i) other crops (ii) geographically different locations.

The details of dataset are not given. Meteorological data important for crop yield prediction. Whether it has been included or not.

The method is not compared with newer DL models.

**Questions:**

As above

---

> ### Author Response · Authors · 2025-11-19
>
> Thank you very much for your careful reading and constructive comments — they helped us substantially improve the manuscript. Below we summarize how we addressed each of your concerns in the revised paper.
>
> We added and discussed relevant prior work on mixed-accuracy (and related) data fusion in the Related Work to better situate our contribution. Concerning the risk that simulated data may bias or be amplified by our procedure, we explicitly tested this in two ways. First, we injected zero-mean Gaussian noise into the low-accuracy (simulated) dataset at multiple magnitudes (0%, 10%, 20%, 30%, 40%) and re-ran training; results (Table 6 and Section 5.4) show that our DDL model continues to outperform the strongest baselines up to 20% noise, and degrades gracefully beyond that level.
>
> To address the lack of interpretability / feature sensitivity analysis, we added a compact analysis in the Appendix (A.2) showing per-sample temporal error patterns and a feature-importance visualization (Figures 5–6).
>
> We expanded empirical evaluation as requested. Section 5.3 now contains cross-species experiments on Wheat and Maize (high/low accuracy sizes reported in Sections 3 and 5.3; corresponding results are in Tables 8–9). We also include a sparse-region generalization experiment (grid partitioning by 1° latitude/longitude) that evaluates geographic robustness; details and results are reported in Section 5.3 and Table 5.
>
> We clarified dataset details and the role of meteorological variables. Section 4.1 now lists all input features used in both datasets (Longitude, Latitude, temperature, precipitation, irrigation, fertilizer N-rate, SOC, clay, pH, bulk density, crop type)
> Finally, to address the comparison with more recent or varied models, we included several additional baselines in the experiments: Gaussian Process Regression (GPR), a Multi-Head-Attention MLP (MHA-MLP), PBA-MLP (as in Ho et al., ICML 2019), and Broad Learning System (BLS) from Liu & Chen (2018). Results for rice (Table 7) and the added cross-species tables (Tables 8–9) show these comparisons; we also added a brief comment in the Results section interpreting where and why DDL outperforms (e.g., better use of mixed accuracy sources) or underperforms (edge cases) relative to particular baselines.
>
> Thank you again for the constructive and actionable feedback — it substantially improved the clarity and empirical rigor of our work.

---

> ### Author Response · Authors · 2025-11-26
>
> Dear Reviewer,
>
> I hope you are having a good week. With the discussion period drawing to a close in one week, we are reaching out to confirm that we have satisfactorily addressed all of your concerns.
>
> Your insights have been invaluable to us, and we are committed to ensuring our paper meets the highest standards. If you have any remaining questions or additional feedback, please do not hesitate to let us know. Thank you again for the time and effort you have dedicated to reviewing our work.

---

### Author Response · Authors · 2025-12-04
**Summary of Revisions: New Baselines, Multi-Crop Evaluation, and Methodological Clarifications**

Dear Area Chair and Reviewers,

We sincerely thank you for the constructive feedback provided during the review process. Guided by your insights, we have significantly revised the manuscript to strengthen the empirical evaluation, clarify the methodological contributions, and improve readability.

All major changes in the updated PDF are **highlighted in yellow**. A summary of the key revisions follows:

**1. Significantly Expanded Empirical Evaluation**
*   **Cross-Species Generalization (Section 5.3 & Appendix A.3):** To address concerns about generalizability beyond rice, we conducted extensive experiments on **Wheat** ($N \approx 53k$) and **Maize** ($N \approx 39k$). DDL consistently outperforms baselines across all three crops (Tables 3, 4, 8, and 9).
*   **New Baselines (Table 2 & Appendix):** We added stronger, architecture-aligned baselines to ensure fair comparison:
    *   **PBA-MLP:** To compare against augmentation scheduling.
    *   **ResNet (Full Data):** To isolate the gain from the architecture vs. the DDL mechanism.
    *   **ResNet (Soft Reweighting):** To demonstrate the superiority of "Hard Discard" over "Soft Reweighting" for handling systematic simulation bias.
*   **Robustness to Noise (Section 5.4):** We added a noise sensitivity analysis (Table 6), proving DDL maintains performance advantages even when simulation data is injected with up to 20% Gaussian noise.
*   **Sparse-Region Generalization (Table 5):** We validated model performance in data-sparse geographical grids.

**2. Theoretical Positioning and Clarity**
*   **Relation to Prior Work (Section 2):** We added a subsection *“Dynamic Reweighting and Data Selection”* to explicitly distinguish DDL from Curriculum Learning and Reweighting (Ren et al., 2018), emphasizing DDL’s unique role in filtering systematic bias in scientific simulations.
*   **Methodological Details (Section 4):**
    *   Corrected the mathematical notation in the Discard Probability formula (Section 4.2.1).
    *   Explicitly listed all 11 input features and defined terms like SOC (Section 4.1).
    *   Clarified the strict train/test split protocol to ensure no data leakage (Section 4.1).
We believe the manuscript is now much stronger and scientifically more rigorous. We are happy to answer any further questions.

Sincerely,
The Authors

---

### Meta-Review · Area_Chair_Y38X · 2026-01-07

**Summary:**

The work addresses an important applied problem, i.e. learning from mixed-accuracy (field vs. simulation) datasets, and the authors made some changes in response to reviews. However, remaining concerns about (i) how distinct the method is from established sample selection/reweighting paradigms, (ii) fairness/completeness of baselines and statistical reporting, and (iii) overstatement/inconsistencies in cross-crop claims have informed this decision.

Across the four reviews, there was, nevertheless, broad agreement that the problem is valuable.

**Reviewer Concerns:**

Some of the issues identified were addressed properly whereas some others remain open.

For instance:

a) Expanded evaluation beyond rice: Authors added experiments on Wheat and Maize.

b) Clarified data protocol and leakage concerns: they clarified a split protocol (hold-out test from high-accuracy data before training; test not used in discard/weighting).

c) Added robustness analysis to noise: They included a noise sensitivity study (injecting Gaussian noise into low-accuracy data), which is relevant given concerns about simulation bias/noise

However open remain issues about:

Novelty remains borderline, and differentiation is still largely empirical/heuristic, including lack of proper baseline comparisons.

**Reviewer Scores:**

Given the original scores and the subsequent rebuttal, I doubt we would have seen any score increases, apart from possibly one reviewer.

---

### Decision · Program_Chairs · 2026-01-26

Reject